# Virtual Screening of Novel 24-Dehydroxysterol Reductase (*DHCR24*) Inhibitors and the Biological Evaluation of Irbesartan in Cholesterol-Lowering Effect

**DOI:** 10.3390/molecules28062643

**Published:** 2023-03-14

**Authors:** Haozhen Wang, Ziyin Lu, Yang Li, Ting Liu, Linlin Zhao, Tianqi Gao, Xiuli Lu, Bing Gao

**Affiliations:** 1The School of Life Science, Liaoning University, Chongshanzhong-Lu No. 66, Shenyang 110036, China; 2Department of Cell Biology and Genetics, Shenyang Medical College, Shenyang 110034, China

**Keywords:** *DHCR24* inhibitor, cholesterol-lowering drug, virtual screening, irbesartan, hyperlipidemia

## Abstract

Hyperlipidemia is a risk factor for the development of fatty liver and cardiovascular diseases such as atherosclerosis and coronary heart disease, and hence, cholesterol-lowering drugs are considered important and effective in preventing cardiovascular diseases. Thus, researchers in the field of new drug development are endeavoring to identify new types of cholesterol-lowering drugs. 3β-hydroxysterol-Δ(24)-reductase (*DHCR24*) catalyzes the conversion of desmosterol to cholesterol, which is the last step in the cholesterol biosynthesis pathway. We speculated that blocking the catalytic activity of *DHCR24* could be a novel therapeutic strategy for treating hyperlipidemia. In the present study, by virtually screening the DrugBank database and performing molecular dynamics simulation analysis, we selected four potential *DHCR24* inhibitor candidates: irbesartan, risperidone, tolvaptan, and conivaptan. All four candidates showed significant cholesterol-lowering activity in HepG2 cells. The experimental mouse model of hyperlipidemia demonstrated that all four candidates improved high blood lipid levels and fat vacuolation in the livers of mice fed with a high-fat diet. In addition, Western blot analysis results suggested that irbesartan reduced cholesterol levels by downregulating the expression of the low-density lipoprotein receptor. Finally, the immune complex activity assay confirmed the inhibitory effect of irbesartan on the enzymatic activity of *DHCR24* with its half-maximal inhibitory concentration (IC50) value of 602 nM. Thus, to the best of our knowledge, this is the first study to report that blocking the enzymatic activity of *DHCR24* via competitive inhibition is a potential strategy for developing new cholesterol-lowering drugs against hyperlipidemia or multiple cancers. Furthermore, considering that irbesartan is currently used to treat hypertension combined with type 2 diabetes, we believe that irbesartan should be a suitable choice for patients with both hypertension and hyperlipidemia.

## 1. Introduction

Cholesterol is an essential biomolecule found in all mammals and plays an important role in cell membrane fluidity and permeability [1]. It is also an important metabolic precursor of vitamin D and bile acid biosynthesis pathways [2]. However, hyperlipidemia, caused by abnormal lipoprotein metabolism leading to an increase in cholesterol, triglyceride, and low-density lipoprotein levels, and a reduction in high-density lipoprotein levels, is an independent and important risk factor for atherosclerotic diseases such as stroke and myocardial infarction in humans. Hyperlipidemia is also directly correlated with serious illnesses such as cancer [3]. Therefore, cholesterol-lowering drugs are considered important and effective for preventing cardiovascular diseases and cancer. Researchers in the field of new drug development are attempting to discover new types of cholesterol-lowering drugs.

The human body produces or obtains cholesterol in two ways: endogenous synthesis and exogenous acquisition. Between these two ways, endogenous cholesterol synthesis is the dominant way. Therefore, blocking the endogenous synthesis pathway is an important strategy for treating hypercholesterolemia. The cholesterol biosynthesis pathway is complex and starts with Acetyl-coenzyme A (acetyl-CoA), and involves more than 30 enzymatic reactions and 20 enzymes that depend on the endoplasmic reticulum. As shown in Appendix A, the conversion of lanosterol to cholesterol involves two different pathways. The Bloch pathway converts desmosterol to cholesterol in the presence of 24-dehydrocholesterol reductase (*DHCR24*), whereas the Kandutsch–Russell pathway requires 7-dehydrocholesterol reductase (DHCR7) to convert 7-dehydrocholesterol to cholesterol [4]. Thus, *DHCR24* is essential for both pathways [5]. It is believed that the enzymatic function of *DHCR24* requests the FAD as a co-factor [6,7].

Statins are the most widely used cholesterol-lowering drugs globally and were first isolated from fungus [8,9]. There are currently seven commonly used statins that inhibit the endogenous synthesis of cholesterol in the liver [10]. Statins are competitive inhibitors of HMG-CoA reductase, which is the rate-limiting enzyme catalyzing the conversion of HMG-CoA to mevalonate [11,12]. Because statins inhibit the synthesis of endogenous cholesterol, the expression of low-density lipoprotein receptors on the surface of hepatocytes increases via the cholesterol homeostatic regulatory mechanism, which leads to the acceleration of plasma low-density lipoprotein decomposition, thereby reducing serum low-density lipoprotein levels [13,14,15]. Although these statins are generally well tolerated, there may be side effects such as myopathy [16], rhabdomyolysis [17], and increased liver transaminase levels [18,19]. Moreover, statins effectively block a series of cholesterol synthesis reactions in the early stages, which considerably reduce the levels of some intermediate products, and some of these intermediate products also play important roles in cellular homeostasis. For example, mevalonate controls various signal transductions such as the Hippo signaling pathways related to cell proliferation [20]. Lanosterol is involved in the Hedgehog signaling pathway that may cause cancer when Hedgehog signaling is incorrectly activated [21].

In the case of cholesterol biosynthesis from lanosterol, the choice of pathway depends on the stage when the double bond on the sterol side chain at C24 is reduced. If the reduction in the C24 double bond is retained until the last reaction, cholesterol synthesis proceeds via cholesta-5,24-dienol (desmosterol) (Bloch pathway) [22]. *DHCR24* catalyzes the reduction in desmosterol to cholesterol, which is the last step in the Bloch pathway of cholesterol biosynthesis. Cholesterol biosynthesis is mainly completed in the liver and involves the Bloch pathway as the main pathway [23]. Among the 10 enzymes involved in distal cholesterol biosynthesis, starting with squalene, *DHCR24* has recently taken a central position in several diseases or pathological conditions. Inhibition of *DHCR24* leads to accumulation of the bioactive metabolite desmosterol, which promotes polyunsaturated fatty acid (PUFA) biosynthesis and production of downstream anti-inflammatory mediators [24]. Moreover, *DHCR24* has been linked to Alzheimer’s disease (AD), oncogenic and oxidative stress [25], hepatitis C virus (HCV) infections [26], differentiation of T helper-17 cells [27], development of foam cells [28], and prostate cancer [29]. Mutation in the *DHCR24* gene in humans causes a decrease in the *DHCR24* enzyme activity, leading to desmosterolosis (MIM 602398), which is a rare and severely autosomal recessive genetic disease [30,31]. Congenital malformations and developmental abnormalities caused by desmosterolosis are attributed to the fetus being unable to effectively obtain sufficient exogenous cholesterol from the mother, and endogenous cholesterol is difficult to synthesize because of the abnormally low activity of *DHCR24*. Triparanol is the first synthetic cholesterol-lowering drug, which was launched in the United States in 1960. The drug acts by inhibiting *DHCR24*, which catalyzes the final step of cholesterol biosynthesis [32]. However, it was withdrawn in 1962 due to serious adverse reactions such as nausea and vomiting, vision loss caused by irreversible cataract, hair loss, skin diseases (such as dryness, itching, peeling and “fish scale” texture), and accelerated atherosclerosis. 3-β-[2-(diethylamino)ethoxy]androst-5-en-17-one (U18666A) [33,34], SH-42 [35], and N,N-dimethyl-3-β-hydroxy-cholenamide (DMHCA) [36] can also inhibit the activity of *DHCR24*, thereby probably reducing cellular cholesterol levels. Compared with the control group, the cholesterol levels in embryonic fibroblasts of *DHCR24* knockout mice significantly reduced, whereas the concentration of desmosterol significantly increased [37]. Desmosterol accumulation to a small extent does not affect vitality in heterozygous carriers of the *DHCR24* mutation, especially when they have a cholesterol-rich diet [38]. Therefore, moderate accumulation of desmosterol by inhibiting *DHCR24* activity has been reported to be non-toxic in mice [35]. These studies suggested that blocking the *DHCR24* activity can be a potential strategy to lower cholesterol levels, thereby reducing blood lipid levels.

In the present study, we constructed and optimized the 3D structure of *DHCR24* on the basis of its amino acid sequence, and used the VS method based on molecular docking to screen candidate compounds that could inhibit *DHCR24* activity. By using molecular dynamics’ simulations and free energy calculations, we further predicted and analyzed the mechanism of interaction between these potential inhibitors and *DHCR24* [39]. Thereafter, we determined the effect of these candidates on lowering cholesterol levels in the human liver cell line HepG2 and an experimental mouse model of hyperlipidemia. Finally, an enzymatic activity assay system of *DHCR24* catalyzing the conversion of desmosterol to cholesterol was constructed in vitro, and the effect of irbesartan on blocking *DHCR24* activity was confirmed.

## 2. Results

### 2.1. DHCR24 Model Building and Its Optimization

The *DHCR24* amino acid sequence (amino acids 23–516) without the signal peptide sequence (amino acids 1–22) was submitted to the I-TASSER server for predicting the 3D structure model of *DHCR24*. The 3D structure model was then optimized with NAMD2.9 at a constant temperature of 300 K for 100 ps (Figure 1A). The Ramachandran plot of *DHCR24* shows that 96.6% of amino acids fall in the favored and allowed regions, indicating that *DHCR24* of a correct model quality was obtained (Figure 1B). Scoring software Maxsub, LGscore, and VERIFY3D were used to evaluate the model. The scores and standard of the correct model are shown in Table 1. The results showed that we could obtain a reasonable 3D structure of *DHCR24*.

### 2.2. Virtual Screening of the DrugBank Database against Desmosterol-Binding Pocket on DHCR24

Before the virtual screening, AutoDock Vina was used for docking the *DHCR24*-FAD complex with desmosterol 5000 times to identify the binding pocket. The docking results showed that the grid box completely contained desmosterol and amino acid residues that interact with it. Based on this result, we determined the active center for virtual screening. After using AutoDock4.2 for accurate docking, 10 small molecules with low binding energy were listed by cluster analysis (Table 2). The selection of drugs was performed based on the drug-likeness judged by the Lipinski rules and whether it was approved by the Food and Drug Administration (FDA). We selected four candidate drugs, namely risperidone (RIS), tolvaptan (TOL), irbesartan (IRB), and conivaptan (CON), for further analysis.

### 2.3. Modes and Binding Energy of Interaction between Four Drug Candidates and DHCR24

We used the molecular dynamics simulation method to further study the dynamic properties of the four candidate drugs and the *DHCR24*-FAD complex. The results were compared with those of the desmosterol-*DHCR24*-FAD complex. The root mean square deviation (RMSD) of the protein–ligand complex were stable after 12 ns (Figure 2A). We then intercepted the balanced conformation and calculated the binding free energy of each system by the Molecular Mechanics–Generalized Born Surface Area (MM-GBSA) method. Compared with the *DHCR24* + FAD + DES system, the *DHCR24* + FAD + IRB or *DHCR24* + FAD + CON system had a more negative binding energy, showing that conivaptan or irbesartan had a better inhibition effect on *DHCR24* (Table 3). To study the binding mode, we intercepted the last frame of the molecular dynamics’ simulation results and aligned the *DHCR24* + FAD + DES system with *DHCR24* + FAD and the four drug-candidate systems, respectively (Figure 2B). The results showed that conivaptan did not bind to the same pocket where desmosterol binds to *DHCR24* but around its active pocket, suggesting that it may play a role in inhibiting the activity of the enzyme by preventing the entry of the physiological substrate desmosterol into the active pocket. The binding position of irbesartan in the *DHCR24* protein structure was the same as the binding position of desmosterol and *DHCR24*. The binding position of risperidone or tolvaptan to the *DHCR24* protein slightly deviated from the binding position of its substrate desmosterol. These results showed that irbesartan may better block the *DHCR24* catalyst activity in cholesterol synthesis through competitive inhibition, whereas the other three candidate inhibitors are slightly less effective.

### 2.4. Atomic Interaction between the Selective Inhibitors and DHCR24

Because conivaptan did not bind to the active pocket of *DHCR24*, intermolecular interaction between each of the other three inhibitors and *DHCR24* was analyzed (Figure 3). The desmosterol binding site of *DHCR24* was a hydrophobic pocket composed of some amino acid residues (Ile383, Leu290, Phe379, Val385, Gln380, Ile229, Leu226, Ser251, Asp382, Asn293, Glu295, and Gly296) on the surface. A hydrogen bond was formed between DES and Leu252 (Figure 3B). The following amino acid residues interacted with risperidone: Tyr299, Leu252, Gly296, Leu226, Glu295, Leu290, Leu376, and Hid377 (Figure 3D). The following amino acid residues interacted with irbesartan: Asp253, Leu252, Tyr250, Ser251, Asn293, Leu376, Gln373, Gly296, Hid377, and Glu295. Two hydrogen bonds were formed between irbesartan and Leu252, and between irbesartan and Hid377. The formation of two hydrogen bonds indicated that IRB and *DHCR24* were more tightly bound compared with desmosterol and *DHCR24*. The following amino acid residues interacted with tolvaptan: Gly296, Glu295, Leu252, Asp253, Ser251, Gln380, Asn381, and Hid377. Our results showed that the binding positions of risperidone, irbesartan, and tolvaptan to *DHCR24* were similar to the desmosterol binding site. The hydrophobic amino acid residues in the three drugs were also approximately similar to desmosterol. Thus, these three drugs have potential roles in the competitive inhibition of desmosterol binding to *DHCR24* and need to be studied further for the best inhibitory effect.

### 2.5. Four Candidate Inhibitors Decrease the Intracellular Cholesterol Levels in HepG2 Cells

After stimulation of HEPG2 cells with the indicated candidate drugs, the filipin staining method was used to determine the cholesterol content of cells. The fluorescence intensity at the cell border was stronger than elsewhere in the cells, indicating that the cholesterol content in the cell membrane was high (Figure 4A,B). This result is also consistent with the widely accepted theory that cell membranes have the highest cholesterol content. Compared with the S (+) and S (−) groups, the fluorescence intensity of the control group (U18666A) and the four drug groups was significantly reduced (*p* < 0.001), indicating that these small molecule drug candidates can inhibit cholesterol synthesis, thereby reducing the content of cholesterol in cells. Image-J was used to analyze the relative fluorescence intensity ratio of each image (Figure 4B). Compared with S (+) and S (−) groups, the cholesterol content in HepG2 cells treated with U18666A and four candidate drugs was significantly reduced, showing that the four drugs obtained by virtual screening exhibited the function of *DHCR24* activity inhibition and cholesterol reduction. After drug treatment of HepG2 cells, the HPLC method was used to detect the changes in cholesterol content (Figure 4C). In all drug-treated groups, the cholesterol content in the cells was reduced to a very low level that was undetectable by HPLC. These data proved that these four drug candidates had a substantial effect on *DHCR24* activity inhibition at the cellular level and cholesterol content reduction.

### 2.6. Four Candidate Drugs Lowered Blood Lipid Levels In Vivo

C57BL/6J mice were fed a high-fat diet to build a mouse hyperlipidemia model. They were bred for 4 weeks, and changes in their weights were recorded weekly (Figure 5A). The analysis of mouse serum parameters showed that the total serum cholesterol (TC) level of mice in the risperidone group and irbesartan group were reduced and the serum LDL-C content of the irbesartan group, conivaptan group, and tolvaptan group were decreased compared with the solvent control group (CMC-Na). The total serum triglyceride (TG) content in all groups was increased, and the serum HDL-C content of all groups was decreased. The IRB group showed a significant lowering in serum total cholesterol (TC), triglycerides (TG), and LDL-C and increasing HDL-C, which is also consistent with the results of molecular dynamics’ simulations (Figure 5B).

The results of HE staining of liver tissues showed that the liver histology of mice in the high-fat model vehicle control CMC-Na group showed larger hepatocytes and lipids aggregated in the cells compared with the liver tissues of normal-diet mice (ND). At the same time, lipids were scattered in the cytoplasm, the nucleus was squeezed to the edge with obvious vacuolation, which proved that the mice fed with a high-fat diet were successfully modeled. Compared with the high-fat group, the irbesartan group, the risperidone group, the tolvaptan group, and the conivaptan group showed that the liver cell volume of mice was smaller with decreased lipid vacuolation. These improved phenotypes were also observed in the simvastatin-treated positive control. These results suggested that all four drugs improved the symptoms of hyperlipidemia in high-fat mice. Based on the results of biochemical indicators in mouse serum, irbesartan performed significantly better among the four drug candidates (Figure 5C).

Because the excellent lipid-lowering effect of irbesartan was observed, we further explored its molecular mechanism. The effect occurs through blocking of the intracellular cellular mevalonate metabolism pathway by inhibiting the endogenous cholesterol synthesis rate-limiting enzyme (HMG-COA reductase) [40]. Thus, the amount and activity of low-density lipoprotein receptor (LDL-R) on the cell surface were increased by a feedback mechanism, and the clearance of cholesterol in the serum increased. Therefore, we speculated that *DHCR24* inhibitors such as irbesartan may also exhibit a lipid-lowering effect through a similar mechanism. The Western blotting results showed that compared with the normal group (ND), the LDL-R expression in the solvent control group with a high-fat diet (CMC-Na) was significantly reduced, whereas the LDL-R level in the positive control group (SM) was significantly increased compared with the solvent control (CMC-Na) (Figure 5D). Similarly, the irbesartan group showed the same effect as the simvastatin group, with an upregulated protein level of LDL-R. This result showed that the mechanism of irbesartan to lower cholesterol is by blocking endogenous cholesterol synthesis and upregulation of LDL-R levels.

### 2.7. Irbesartan Inhibits DHCR24 Enzyme Activity

Finally, we confirmed the direct effect of irbesartan on enzymatic activity inhibition of *DHCR24* in the conversion of desmosterol to cholesterol by performing the immune complex-activity assay. To obtain a large amount of *DHCR24* enzyme, we infected HepG2 cells with recombinant adenovirus Ad-cmv-*DHCR24* and performed immunoprecipitation using an IP antibody against *DHCR24*. Western blotting analysis of protein extracts from adenovirus-infected HepG2 cells revealed that *DHCR24* protein was expressed in large amounts (Figure 6A). The antibody-*DHCR24* enzyme complex (IP complex) was finally obtained from the HepG2 cells infected with adenovirus over-expressing *DHCR24* after immunoprecipitation. As shown in Figure 6B, no band was detected in the IP complex of the Ad-LacZ control group, whereas the *DHCR24* protein was expressed in a large amount in the Ad-cmv-*DHCR24* infected group, indicating that we successfully obtained the antibody-*DHCR24* enzyme complex for determining the enzymatic activity of *DHCR24*.

In the immune complex-activity assay of *DHCR24*, the HPLC method was used to detect desmosterol and cholesterol in the assay mixture. Under the HPLC assay conditions, desmosterol was detected in about 25 min, and cholesterol was detected in about 35 min (a–c in Figure 6C). To test the inhibitory effect of irbesartan on *DHCR24* enzymatic activity in vitro, we constructed a cholesterol synthesis system catalyzed by the *DHCR24* enzyme in vitro. The system is composed of substrate desmosterol, EDTA, DTT, NAD, NADP, and FAD. The *DHCR24* protein required for the system was included in the antibody-*DHCR24* enzyme complex after IP using the cell lysates of Ad-CMV-*DHCR24*-infected HepG2 cells. As shown in d–f of Figure 6C, desmosterol in the system can be catalyzed into cholesterol only in the presence of the *DHCR24* enzyme. The remaining desmosterol was high but cholesterol was undetectable in the reaction mixture with U18666A. This suggested that U18666A could inhibit *DHCR24* enzymatic activity in the reaction system (in e and f of Figure 6C). To further confirm the cholesterol-lowering effect of irbesartan, different concentrations of irbesartan were used (0 nM, 200 nM, 500 nM, 1 μM, and 2 μM), which were added to the cholesterol-synthesis system in vitro. We found that the content of cholesterol in the IRB group decreased in a dose-dependent manner; the data of 1 mM IRB are shown in h of Figure 6C. Finally, IC50 was calculated by plotting the graph between different concentrations of irbesartan and % inhibition of *DHCR24* activity, which was found to be 602 nM (Figure 6D).

## 3. Discussion

*DHCR24* is an enzyme that catalyzes the formation of cholesterol. The inhibition of its activity is an effective strategy for hyperlipidemia treatment. First, we built and optimized the three-dimensional structure of *DHCR24* (Figure 1 and Table 1). We then identified the binding pocket for desmosterol binding to *DHCR24* and used it as the active center for virtual screening. The results of virtual screening and molecular dynamics simulation showed that irbesartan, risperidone, tolvaptan, and conivaptan are possible *DHCR24* inhibitors (Table 2 and Table 3, Figure 2 and Figure 3). Filipin staining and HPLC analysis at the cellular level showed that all four drugs exhibited a cholesterol synthesis inhibitory effect (Figure 4). The analysis of serum parameters of mice showed that all four drugs reduced cholesterol production (Figure 5A,B). They also reduced mouse liver cell volume and lipid vacuolation (Figure 5C). Irbesartan was the most effective among the four drugs, which is consistent with the results of virtual screening and molecular dynamics simulation. We then proved that irbesartan decreases cholesterol production via the downregulation of LDL-R expression (Figure 5D). We used IP and HPLC methods to detect the irbesartan-induced inhibition of *DHCR24* enzyme activity in vitro with an IC_50_ value of 602 nM (Figure 6).

As *DHCR24* catalyzes the last step of cholesterol formation, the inhibition of its activity reduced the synthesis of endogenous cholesterol. Mutations in the *DHCR24* gene can cause the accumulation of desmosterol, and the clinical observations of patients who were heterozygous for the *DHCR24* mutation showed that moderate desmosterol accumulation was non-toxic to humans. Thus, reducing cholesterol by inhibiting the activity of *DHCR24* is feasible, and no *DHCR24* inhibitor in the market is a cholesterol-lowering drug. In addition, a study showed that inhibiting the expression and activity of *DHCR24*, as well as cholesterol biosynthesis and the formation of lipid rafts mediated by *DHCR24*, can inhibit the invasion and migration of hepatocellular carcinomas [41]. Therefore, targeting *DHCR24*-mediated cholesterol metabolism may be an effective treatment strategy for liver cancer.

Irbesartan is currently used to treat hypertension with type 2 diabetes and has a protective effect on the kidneys. Hence, irbesartan may be a better alternative for patients with both hypertension and hyperlipidemia. Clinical data showed that irbesartan can reduce TC, TG, and LDL-C and increase HDL-C in patients with both hypertension and hypercholesterolemia without an exact mechanism [42]. Our results showed that irbesartan can block endogenous cholesterol synthesis by inhibiting *DHCR24* activity and can increase LDL-R levels to reduce cholesterol levels. In previous studies, risperidone could slightly reduce desmosterol accumulation in D7-N2A cells, and conivaptan and tolvaptan could elevate desmosterol levels with unclear mechanisms [43]. Our results provide a new explanation that conivaptan and tolvaptan could elevate the desmosterol level though probably block the cholesterol biosynthesis by inhibiting *DHCR24*′s activity. Although these drugs were not as effective as irbesartan in lowering cellular cholesterol, they significantly improved fat vacuolation in mice compared with the high-fat diet control group in the present animal experiments.

This study only verified the cholesterol-lowering effects of four *DHCR24* inhibitors through cell lines and animal experiments. Our results do not clearly explain the mechanism of irbesartan’s effect on cholesterol lowering, however, we believe that irbesartan might block the intracellular mevalonate metabolism pathway of cholesterol biosynthesis by inhibiting enzymatic function of *DHCR24* and upregulate the amount and activity of LDL-R on the cell surface. The key amino acid residues involved in irbesartan binding to *DHCR24* have not been specifically predicted or verified. Although the inhibitory effect of irbesartan on *DHCR24* is better than that of U18666A, it is still inferior to another experimental *DHCR24* inhibitor SH-42 (IC_50_ = 4 nM) [35]. In addition, selecting more known *DHCR24* inhibitors as the control during screening may lead to more potential drug candidates. They should be studied in further experiments.

To summarize, our experimental results proved that drug repurposing is desirable in drug development and irbesartan might be considered a new drug for hyperlipidemia.

## 4. Materials and Methods

### 4.1. Construction of 3D Structure of DHCR24 Proteins

The amino acid sequence of *DHCR24* was retrieved and downloaded from the UniProt databank (Accession Number: Q15392) (http://www.uniprot.org/, accessed on 1 September 2018) that reported the location of the signal peptide cleavage site between Gly22 and Leu23 [44]. After deleting the signal peptide sequence, we submitted the sequence containing 494 amino acids to the I-TASSER Server in order to construct the 3D structure of the *DHCR24* protein [45,46]. I-TASSER (Iterative Threading ASSEmbly Refinement) is a hierarchical approach to protein structure and function prediction. It first identifies structural templates from the PDB by multiple threading approach, with full-length atomic models constructed by iterative template fragment assembly simulation. I-TASSER only uses the templates of the highest significance in the threading alignments, the significance of which are measured by the Z-score, i.e., the difference between the raw and average scores in the unit of standard deviation. Z-score > 1 means a good alignment. The Z-scores of templates selected from the LOMETS threading programs for the *DHCR24* are all greater than 1. NAMD2.9 [47,48] was used to optimize the structure of *DHCR24* at a 300 K constant temperature and to perform molecular dynamics simulations for 100 ps. Ramachandran plot [49] and Scoring software Maxsub [50], LGscore [51], and VERIFY3D [52] were used to evaluate the model.

### 4.2. Preparation of Data Set

The structures of substrate desmosterol (ChemSpider ID: 388662) and the cofactor FAD (ChemSpider ID: 559059) were downloaded from ChemSpider (http://www.chemspider.com/, accessed on 1 September 2018). Then, Pymol [53] was used to change the mol format to the PDBQT. FAD and desmosterol were sequentially docked with *DHCR24* to determine the desmosterol-binding pocket as the active site.

The DrugBank database was used to screen potential inhibitors that bind to the desmosterol-binding site of *DHCR24*. The latest release of DrugBank (version 5.1.10, released 4 January 2023) contains 15,441 drug entries, including 2739 approved small-molecule drugs, 1575 approved biologics (proteins, peptides, vaccines, and allergenics), 134 nutraceuticals, and >6716 experimental drugs. At the time of the screening process, 6858 drugs possessed 3D structures for screening and the 3D structures of these molecules were obtained from DrugBank (https://www.drugbank.ca/, accessed on 1 September 2018) [54]. Then, we used OpenBabel2.3 to convert chemical compounds from SDF to the PDBQT, which is required by the AutoDock Vina1.2.0. Meanwhile, the charges were added to each compound and split into individual files.

### 4.3. Virtual Screening

First, the docking software AutoDock Vina1.2.0. [55], based on the Lamarckian Genetic Algorithm, was used to dock the small molecules into the active site of the *DHCR24*_FAD complex 5000 times. The grid box was determined by aligning the structure of the *DHCR24*_FAD_des complex with the structure of desmosterol, and the x, y, and z-coordinates of the grid box (x = 72.193, y = 51.112, z = 24.362) were determined, with the grid point spacing set at 1 Å. A total of 50 potential inhibitors were selected based on the lowest binding energy. Then, the docking software AutoDock 4.2 [56,57] was used to dock the 50 small molecules with low binding energy into the active site of the *DHCR24*_FAD complex. The grid affinity maps were calculated using AutoGrid4.2. Ultimate potential inhibitors were selected based on the binding mode of the inhibitor, and the estimated binding energy of the lowest energy conformation of the highest populated cluster.

### 4.4. Molecular Dynamic Simulations

After exact docking, the LEaP of AmberTools18 [58] was used to create topology and coordinate files for the simulations of the protein–ligand complexes. Four docking complexes of potential inhibitors with *DHCR24*_FAD and a structure of *DHCR24*_FAD_des were constructed for molecular dynamics (MD) simulation to examine whether the candidate compounds could stably bind to the desmosterol binding site of *DHCR24*_FAD as well as to analyze the interaction between potential inhibitors and *DHCR24*_FAD [59,60]. All MD simulations were performed using the NAMD version 2.9 with the ff12SB protein force field and the general AMBER force field (GAFF). Each complex was immersed in the TIP3P water box (12 Å from the solute surface) and neutralized by the addition of Na+ or Cl^-^ ions. Initially, each system was subjected to energy minimization for 30,000 steps. Thereafter, the systems were sequentially heated up from 0 to 300.0 K. Finally, the non-constrained production simulations were performed for 30 ns in the constant temperature and constant pressure (NPT ensemble) simulation [61,62,63].

The root mean square deviation (RMSD) value was the average value of the movement offset of each skeleton C atom of the protein in the system, and the stability of the system could be judged by monitoring its value. The MD simulations were monitored by examination of the internal energy and RMSD of the resulting trajectories. The Visual Molecular Dynamics (VMD) program1.9.4 was employed to animate and analyze the trajectory of MD simulation [64].

### 4.5. MM-GBSA Calculation

We set interval = 1 to collect the last 2000 snapshots from the production stage of MD simulations to calculate the binding free energy of each potential inhibitor or desmosterol to the *DHCR24*_FAD complex by Molecular Mechanics–Generalized Born Surface Area (MM-GBSA) method. This method employs the calculated molecular mechanics’ energies and implicit solvation models to compute the difference between the energy of the bound complex (protein–ligand) and the energies of the unbound protein (protein) and the isolated molecular (ligand) at a reasonable computational cost using the NAMD package [65,66]. Therefore, the TIP3P water model was not included in the MM-GBSA free-energy calculation results.
∆Gbind = <Gcomplex> − <Gprotein> − <Gligand>(1)

This binding free energy can be calculated as described in Equations (2)–(5):∆Gbind = ∆Ggas + ∆Gsol(2)
∆Ggas = ∆Eele + ∆Evdw(3)
∆Gsol= ∆Gpolar + ∆Gnonpolar(4)
∆Gnonpolar =γSASA + β(5)

Furthermore, ∆Ggas corresponds to the sum of electrostatic (∆Eele), van der Waals (∆EvdW), and internal (∆Einternal) energies. The polar solvation free energy (∆Gpolar) was calculated using the generalized Born implicit solvent model. The non-polar solvation free energy (∆Gnonpolar) was calculated by using the solvent-accessible surface area (SASA) algorithm.

### 4.6. Cell Culture

Human hepatocellular carcinoma cell lines (HepG2, Database Name: ATCC, accession Number: HB-8065) were cultured in DMEM supplemented with 10% (*v*/*v*) fetal bovine serum (FBS) and antibiotics (100 μg/mL penicillin G and 100 μg/mL streptomycin) under 5% CO_2_ at 37 °C before treatment. The cells were incubated in media containing risperidone, tolvaptan, irbesartan (Meilunbio, Dalian, China), or conivaptan (Abmole, Shanghai, China) when they reached 60% confluence, and the cells were cultured in DMEM with or without FBS as a control for 24 h.

### 4.7. Detection of Intracellular Cholesterol by Filipin Staining and HPLC

Filipin staining is a widely used fluorescent staining method for cholesterol determination. It can mark free cholesterol and esterified cholesterol on the structure of biofilm. The treated HepG2 cells were transferred to glass slides, stained with filipin reagent, and immediately observed under a fluorescence microscope. The cholesterol deposits exhibited blue fluorescence after staining.

The high-performance liquid phase (HPLC, SHIMADZU, Kyoto, Japan) method was used to determine the effect of four candidate drugs on the intracellular cholesterol content after extracting the intracellular lipid contents. C18 is used as chromatographic column and methanol (chromatographic grade) is used as mobile phase at 25 °C. The flow rate was set to 1.0 mL/min and the drugs were analyzed by UV spectroscopy (SHIMADZU, Kyoto, Japan) at 210 nm.

### 4.8. Mice and Treatments

Male C57BL/6J mice were purchased from Changsheng Bio-technology (Jilin, China) and housed under standard conditions with access to food and water in a temperature-controlled environment. The feeding experiments were grouped into 7 groups on average. The first group included 12 C57BL/6J mice (ND) fed with a normal diet supplemented with CMC-Na, which is used to dissolve drugs. Other mice groups were fed with a high-fat diet with or without the supplementation of 1%CMC-Na, simvastatin (20 mg/kg), risperidone (0.7 mg/kg), tolvaptan (1 mg/kg), irbesartan (6 mg/kg), or conivaptan (1 mg/kg).

The mice received the prescribed diets and supplementation for 4 weeks. At the end of the feeding period, the mice were sacrificed by exsanguination under diethyl ether. Their livers were excised immediately, and the serum was separated from the blood for analysis.

### 4.9. Analysis of the Liver Tissues and Serum Parameters

The collected livers were fixed in 4% paraformaldehyde and embedded in paraffin. The sections obtained were stained with hematoxylin and eosin. The triacylglycerol, cholesterol, low-density lipoprotein cholesterol (LDL-C), and high-density lipoprotein cholesterol (HDL-C) levels in the serum were measured using enzyme assay kits from NanJing JianCheng Bioengineering Institute (Nanjing, China).

### 4.10. Adenovirus-Mediated Overexpression of DHCR24

We previously constructed an adenovirus (Ad)-cmv-*DHCR24* recombinant adenovirus [67]. The HepG2 cells were infected with Ad-cmv-*DHCR24* or Ad-LacZ in 2 groups for 72 h, and a serum-free medium was added to the other group serving as control. Each group of cells was centrifuged at low speed, washed twice at 4 °C in cold PBS, and resuspended in RIPA lysis buffer and PMSF (99:1). After incubation on an ice bath for 30 min, the cells were gently homogenized. The cell lysates were then extracted from each group of cells for further analysis. A small portion of the cell lysates was transferred to new microtubes for Western blotting to determine the expression levels of *DHCR24* in the total cell lysates.

### 4.11. Western Blotting

The proteins were extracted from HepG2 cells and the liver tissues with RIPA lysis buffer containing PMSF, while the protein concentration was determined according to the instructions of the Bicinchoninic Acid (BCA) kit (Thermo Scientific, Shenyang, China). The samples were mixed with 5* loading buffer and heated on a metal bath at 95 °C for 5 min to denature the proteins. The proteins were then separated by 10% sodium dodecyl–sulfate polyacrylamide gel electrophoresis and transferred onto a nitrocellulose membrane. The membrane was then blocked and blotted with primary antibodies at 4 °C overnight, followed by treatment with horseradish peroxidase (HRP)-conjugated secondary antibody (1:5000) for 1.5 h at 28 °C. Then, the protein bands were visualized using a chemiluminescence detection procedure (DNR Bio-imaging, Jerusalem, Israel). The band intensity was quantified using ImageJ (software https://imagej.nih.gov/ij/, accessed on 5 May 2019). The primary antibodies used in this experiment were as follows: anti-*DHCR24* (Santa Cruz Biotechnology, Inc., Santa Cruz, CA, USA) and anti-β-actin (Thermo Scientific, Shanghai, China).

### 4.12. Immunoprecipitation

*DHCR24* was captured from the infected HepG2 cells with an anti-*DHCR24* antibody (Santa Cruz Biotechnology, Santa Cruz, CA, USA). The required amount of protein A beads was dispensed into microtubes with a wide-bore pipette tip, and the beads were washed with cold PBS. Then, 1 mL of cold PBS was added and mixed, followed by spinning the beads at 3000 rpm in a microcentrifuge for 1 min and removal of the supernatant. This step was repeated thrice. The samples were incubated at 4 °C for 1.5 h with gentle rotation for binding of the immune complexes to protein A-agarose beads. The resultant mixture was then centrifuged at 3000 rpm at 4 °C for 15 min, and the pellet was washed twice in lysis buffer and re-centrifuged. After the last centrifugation step, the protein complexes were eluted from the beads by gentle washing thrice.

### 4.13. DHCR24 Enzyme-Activity Measurements

We created a cholesterol synthesis system in vitro containing 100 mM Tris/HCl (pH 7.23), 0.1 mM EDTA, 1 mM DTT, 30 mM nicotinamide, 3.5 mM nicotinamide adenine dinucleotide phosphate (NADP), 30 mM glucose-6-phosphate, 2 U glucose-6-phosphate dehydrogenase/mL, 0.5 mg bovine serum albumin/mL, 20 μM FAD, and 168 mM desmosterol [6]. The *DHCR24* activity was measured by incubation of 25 μL of the resulting antibody-*DHCR24* complex in 225 μL of the assay mix for 4 h at 37 °C.

For examination of the effect that irbesartan has on the *DHCR24* activity, different concentrations of irbesartan (0 nM, 200 nM, 500 nM, 1000 nM, 2000 nM) were added in the assay mix, as described earlier.

The samples were analyzed by HPLC (HITACHI, Tokyo, Japan), operated at 1 mL/min for 35 min, and analyzed by UV spectroscopy at 210 nm. The sample desmosterol and cholesterol peak areas (mAU·s) obtained from HPLC analysis were used to compare the effect of irbesartan at different concentrations on the *DHCR24* activity. Nonlinear regression was performed by plotting the peak areas vs. the concentration using the GraphPad Prism software9. *DHCR24* inhibition was calculated as follows:Inhibition % = (C_0_ − C*_i_*)/ C_0_⋅100%(6)
where C_0_ means the cholesterol concentration without irbesartan and C*_i_* is the cholesterol concentration with inhibitor. The inhibitory activities of irbesartan were evaluated by IC_50_ values [40]. A dose–inhibition curve was first constructed from the experimental data.

## Figures and Tables

**Figure 1 molecules-28-02643-f001:**
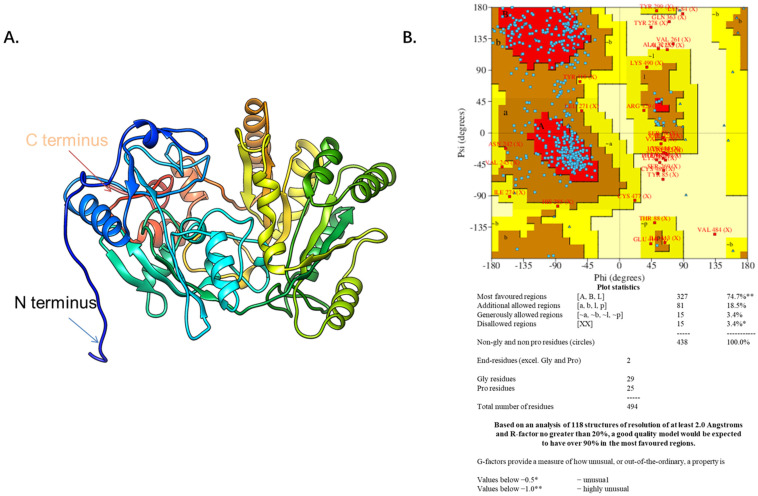
The predicted *DHCR24* structure and evaluation. (**A**). The *DHCR24* model. (**B**). The Ramachandran plot of modeled structure was validated by the PROCHECK program. The red area indicates the most favored regions of amino acid residues, the brown area indicates the additional allowed regions, and the yellow area indicates the generously allowed regions. The Ramachandran plot of protein contained 96.6% of residues in the favored and allowed regions, indicating a good quality model for the protein.

**Figure 2 molecules-28-02643-f002:**
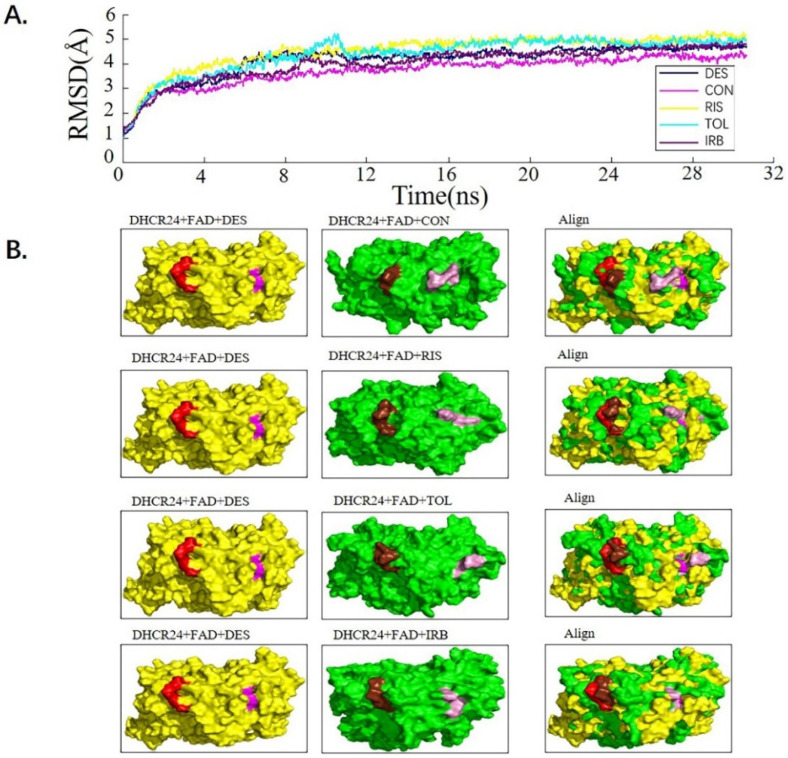
Analysis of desmosterol and inhibitors binding to *DHCR24* through molecular dynamics. (**A**) Plots of Root Means Squared Deviation (RMSD) values during 30 ns in the simulation times corresponding to molecular dynamics of all complexes under study. (**B**) The alignment of 4-docked candidate inhibitors to desmosterol in the complex with *DHCR24*. *DHCR24* is shown as the yellow or green surface, desmosterol as the red surface, FAD as the chocolate surface, and the candidate inhibitors as the pink surface. DES is desmosterol; FAD is flavin adenine dinucleotide; CON is conivaptan; RIS is risperidone; TOL is tolvaptan; IRB is irbesartan.

**Figure 3 molecules-28-02643-f003:**
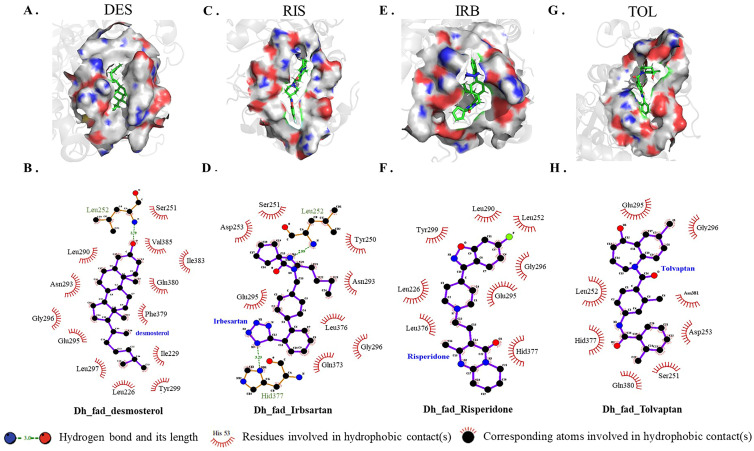
Atomic interaction of desmosterol and three inhibitors with *DHCR24*. (**A**,**C**,**E**,**G**) Hydrogen and hydrophobic interactions as plotted by LIGPLOT. Desmosterol, irbesartan, risperidone and tolvaptan are indicated in blue font. (**B**,**D**,**F**,**H**) The key amino acid residues(green) participated in the binding interaction with inhibitors. DES is desmosterol; RIS is risperidone; IRB is irbesartan; TOL is tolvaptan.

**Figure 4 molecules-28-02643-f004:**
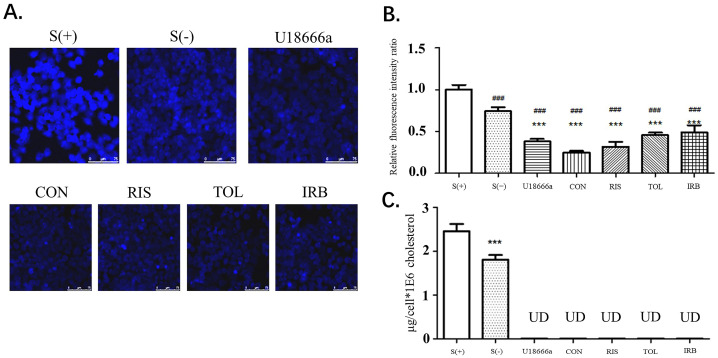
Four candidates inhibited the decrease in the intracellular cholesterol levels of HepG2 cells. (**A**). HepG2 cells were fixed with 4% formaldehyde and treated with filipin at 37 °C for 2 h. After washing with PBS, the cells were viewed by fluorescent microscopy. CON is conivaptan; RIS is risperidone; TOL is tolvaptan; IRB is irbesartan. (**B**). Fluorescence determined using the Image-J software. Data are expressed as means ± SD. T-test and ANOVAs were used for the comparison of two groups and multiple groups, respectively (*n* = 30 for each group). ###: *p* < 0.01 vs. S(+),***: *p* < 0.01 vs. S(−). (**C**). HPLC analysis of the intracellular cholesterol content. UD: undetectable *n* = 3, mean ±SD, ***: *p* < 0.01 vs. S (+).

**Figure 5 molecules-28-02643-f005:**
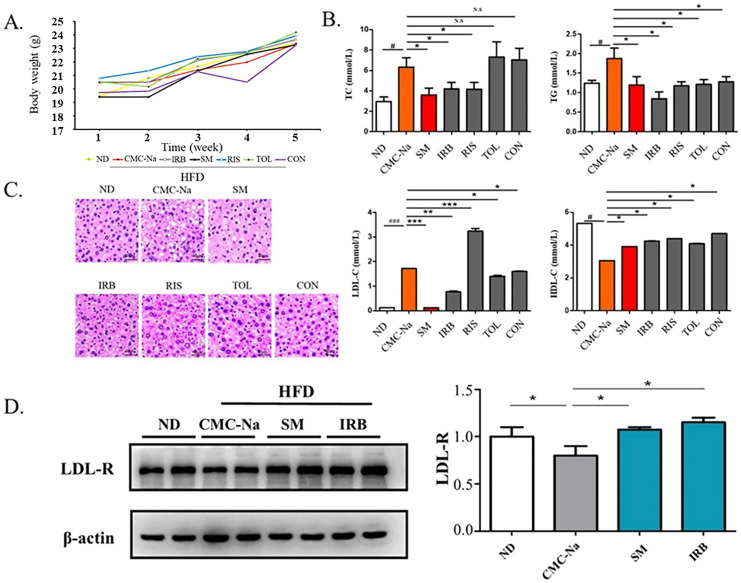
Four candidate inhibitors lowered the blood lipid content in rats through the upregulation of the expression of LDL receptors. (**A**) Bodyweight in grams over 4 weeks of mice on ND and HFD diets: ND is a normal diet; HFD is a high fat diet; CMC-Na is sodium carboxymethyl cellulose; IRB is irbesartan; SM is simvastatin; RIS is risperidone; TOL is tolvaptan; CON is conivaptan. (**B**) The serum levels of TC, TG, LDL-c, and HDL-c(mmol/L) of mice fed with ND or HFD diet (data are mean ± SD, *n* = 12). Significance symbols are denoted as # *p* < 0.05, ###: *p* < 0.01, ND vs. CMC-Na/HFD; * *p* < 0.05, vs. CMC-Na/HFD; ** *p* < 0.01 vs. CMC-Na/HFD; *** *p* < 0.001 vs. CMC-Na/HFD; N.S = no significance. (**C**) Representative images showing hematoxylin and eosin (HE) staining of the liver tissues. Scale bars, 40 µm. (**D**) Western blots indicating the amounts of LDL-R and β-actin in the liver tissues collected from mice. Data presented as mean ± S.D., *n* = 2. * *p* < 0.05 (ANOVA).

**Figure 6 molecules-28-02643-f006:**
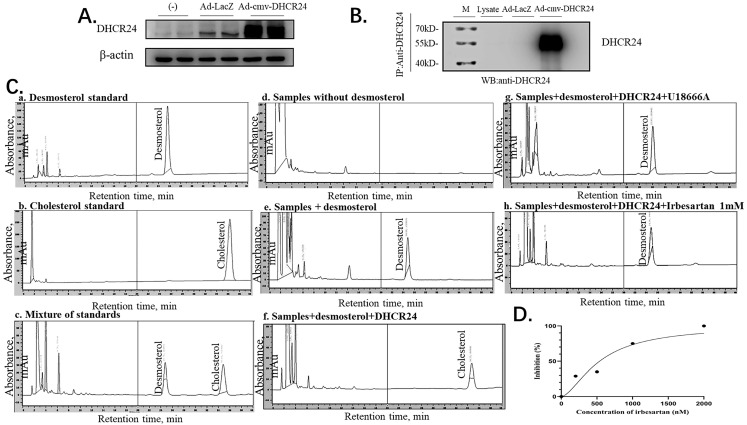
Irbesartan inhibits cholesterol synthesis in the immune complex-activity assay. (**A**) Western blotting indicates the amounts of β-actin and adenovirus-mediated overexpression of *DHCR24*. (**B**) Western blots indicating the amounts of immunoprecipitated *DHCR24* and β-actin. (**C**) HPLC chromatogram of desmosterol and cholesterol. The chromatographic conditions used were as follows: column, C18; mobile phase, methanol; flow-rate, 1 mL/min; detection, ultraviolet absorbance at 210 nm. (**D**) Correlation of the inhibition (%) of *DHCR24* and the concentration of irbesartan (nM) R^2^ = 0.9344.

**Table 1 molecules-28-02643-t001:** LGscore, Maxsub, VERIFY3D, and Ramachandran plot score of the *DHCR24* models.

Score Software	Standard of Correct Model	Score
LGscore	>1.5 correct model	2.53
Maxsub	>0.1 correct model	0.17
VERIFY3D	80% of the residues had an averaged 3D-1D score >= 0.2	82.79%
Ramachandran plot	favored and allowed region > 90%)	96.6%

**Table 2 molecules-28-02643-t002:** The hit compounds by virtual screening and its inhibitory activity against *DHCR24*.

Compound	DrugBank ID	Drug Name	Binding Energy (Kcal/mol)	Drug Types	Drug-Likeness
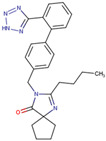	DB01029	Irbesartan	−7.13	Approved, Investigational	Yes; 1 violation: MLOGP > 4.15
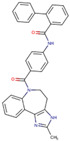	DB00872	Conivaptan	−7.03	Approved, Investigational	Yes; 0 violation
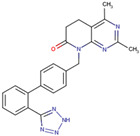	DB01349	Tasosartan	−6.65	Experimental	Yes; 0 violation
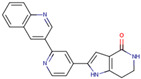	DB08358		−6	Experimental	Yes; 0 violation
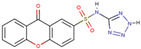	DB04698		−5.91	Experimental	Yes; 0 violation
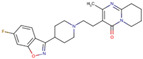	DB00734	Risperidone	−5.79	Approved, investigational	Yes; 0 violation
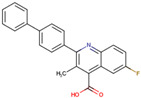	DB03480	Brequinar Analog	−5.57	Experimental	Yes; 0 violation
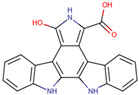	DB07241		−5.5	Experimental	Yes; 0 violation
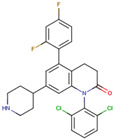	DB01948		−5.19	Experimental	Yes; 1 violation: MLOGP > 4.15
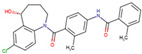	DB06212	Tolvaptan	−5.06	Approved	Yes; 1 violation: MLOGP > 4.15

**Table 3 molecules-28-02643-t003:** The binding affinity of complexes resulting from MM-GBSA analysis with desmosterol or hits compounds.

Energy Component	*DHCR24*-FAD- Desmosterol	*DHCR24*-FAD-Conivaptan	*DHCR24*-FAD- Risperidone	*DHCR24*-FAD-Tolvaptan	*DHCR24*-FAD- Irbesartan
ΔEvdW	−38.06	−34.07	−31.05	−29.47	−44.09
ΔEele	−2.07	−140.17	−1.65	−12.10	−20.84
ΔGpolar	15.77	145.68	13.39	25.38	37.38
ΔGnonpolar	−4.92	−4.65	−3.71	−3.85	−5.09
ΔGgas	−40.13	−174.24	−32.71	−41.58	−64.93
ΔGsol	10.85	141.03	9.68	21.53	32.29
ΔGbind	−29.28	−33.21	−23.03	−20.05	−32.64

DES is desmosterol; FAD is flavin adenine dinucleotide; CON is conivaptan; RIS is risperidone; TOL is tolvaptan; IRB is irbesartan. ∆EvdW is van der Waals energy; ∆Eele electrostatic energy, ∆Gpolar is polar solvation free energy; ∆Gnonpolar is non-polar solvation free energy; ΔGgas is a sum of van der Waals and electrostatic energies; ∆Gsol is solvation free energy; ΔGbind is binding free energy.

## Data Availability

Not applicable.

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
