# Peer review of "Virtual Screening of Novel 24-Dehydroxysterol Reductase (DHCR24) Inhibitors and the Biological Evaluation of Irbesartan in Cholesterol-Lowering Effect"

_molecules, 2023, doi:10.3390/molecules28062643_

Round 1
Reviewer 1 Report
The authors identified a group of drugs as potential DHCR24 inhibitors by virtual screening assays. The proof of concept was carried out by in vitro and in vivo testing. The topic of DHCR24 and the modulation of cholesterol biosynthesis precursors is very exciting as a new target for drug development. Hence, the findings of the authors that irbesartan inhibits DHCR24 is very interesting. Irbesartan could be used as a lead structure for the development of novel DHCR24 inhibitors. I do not believe the conclusion drawn by the authors that a combination therapy for the treatment of hypercholesterolemia and hypertension with irbesartan is useful.
However, the article is not suitable for publication in Molecules because the presented data are insufficient, experiments are missing or described insufficent, and also the handling of the literature could be even better.
Major remarks:
· L29: Was the IC50 value determined so accuratel, that the use of decimal places is meaningful? What was the confidence interval, R2 and how many replicates were used? I would suggest to use a sigmoidal dose response curve and I also recommend to compare the IC50 of irbesartan to a reference inhibitor like DMHCA, SH42 (or triparanol).
· L79-106: The story of triparanol is completely missing. Triparanol was the first market inhibitor of DHCR24, with several side effects, which led to a withdrawal and led to a bad image of DHCR24 inhibitors. Side effects like myotonia and cataracts should be mentioned (doi: 10.2174/0929867328666211115121832.) which are probably substance specific.
· L96: U18666A is not a selective inhibitor of DHCR24
· L79-106: The article of Nes should be mentioned (doi.org/10.1021/cr200021m)
· L81: The favoured biosynthesis is cell type and tissue specific
· L120-206: Why was FAD used as a co-factor? In literature (see also keg pathway and reactome) NADP is used? Recommend on this! I also suggest to use a reference inhibitor as a prove concept. Perhaps one compound from Ned Porter et al. or triparnol. I assume the binding of steroidal DHCR24 inhibitors is different to non-sterol inhibitors. What about verapamil, zolpidem, …
· L208-225: Is the staining with Filipin highly selective? I can imagine that also desmosterol could be stained. Please provide references or experiments or even U18666.
· L214: I would suggest to use another inhibitor U18666 is a dirty drug.
· L376-378: It’s too speculative, especially that irbesartan can be used for cancer treatment. The IC50 value of 600 nM on the isolated enzyme is quite high. I assume there are more potent inhibitors available.
· L473-477: The used method should be described in more detail or a reference should be given. Which column was used, mobile phase, temperature, injection volume, …
· L478-489: What dosage was used
· L497: “ We previously constructed …” reference is missing
· L502-505: “were then extracted” which procedure was used?
Minor remarks:
· L34: “hypertension and hyperlipidemia” it is speculative
· L63-64: Delete the sentence
· L104: “in vivo” I would suggest in mice because it was not determined in man
· L150-169: The abbreviations should be explained when they are used the first time
· L208: HepG2 cells
· L241: What means CMC-Na?
· L274: rats or mice?
· Figure 6: Enlarge the chromatograms. What is shown by the bar around 15min? Chromatograms of references inhibitors should be added
· L470: What means “free cholesterol and cholesterol”? Cholesterol is always free cholesterol
Author Response
Answer to reviewer1:
Comments:
The authors identified a group of drugs as potential DHCR24 inhibitors by virtual screening assays. The proof of concept was carried out by in vitro and in vivo testing. The topic of DHCR24 and the modulation of cholesterol biosynthesis precursors is very exciting as a new target for drug development. Hence, the findings of the authors that irbesartan inhibits DHCR24 is very interesting. Irbesartan could be used as a lead structure for the development of novel DHCR24 inhibitors. I do not believe the conclusion drawn by the authors that a combination therapy for the treatment of hypercholesterolemia and hypertension with irbesartan is useful.
However, the article is not suitable for publication in Molecules because the presented data are insufficient, experiments are missing or described insufficent, and also the handling of the literature could be even better.
We appreciate your valuable suggestions represented in the major and minor concerns below. We have revised the manuscript and answered your questions very carefully. We hope the quality of our revised manuscript is significantly improved according to your suggestions. Thank you again for your important comments.
Major remarks
1) L29: Was the IC50 value determined so accuratel, that the use of decimal places is meaningful? What was the confidence interval, R2 and how many replicates were used? I would suggest to use a sigmoidal dose response curve and I also recommend to compare the IC50 of irbesartan to a reference inhibitor like DMHCA, SH42 (or triparanol).
We appreciate your important suggestions. We agree that for IC50, the decimal places are usually not used in either nM or μM units. Therefore, we deleted the decimal places of IC50 in the manuscript (L27, L316, L340). In this study, different concentrations of irbesartan (0 nM, 200 nM, 500 nM, 1000 nM, 2000nM) were added into the system and its inhibitory effect on DHCR24 was tested by HPLC thrice.
We used GraphPad Prism9.0 software to redraw the sigmoidal dose response curve of IC50 (L569-571) and added into Fig. 6D. According to the relationship between correlation of the inhibition (%) of DHCR24 and the concentration of irbesartan (nM) calculated by nonlinear regression method, R2 = 0.9344 which is > 0.8(L338-339). We also agree with you about using other DHCR24 inhibitors such as SH-42(IC50=42nM) and irbesartan as a comparison, and we added the comparison results to the discussion in the revised manuscript (L389-393).
2) L79-106: The story of triparanol is completely missing. Triparanol was the first market inhibitor of DHCR24, with several side effects, which led to a withdrawal and led to a bad image of DHCR24 inhibitors. Side effects like myotonia and cataracts should be mentioned (doi: 10.2174/0929867328666211115121832.) which are probably substance specific.
Thank you very much for your valuable suggestions and reminder. We added the story of triparanol to our revised manuscript. Triparanol is the first synthetic cholesterol-lowering drug, which was launched in the United States in 1960. The drug acts by inhibiting DHCR24, which catalyzes the final step of cholesterol biosynthesis (doi: 10.2174/0929867328666211115121832). However, it was withdrawn in 1962 due to serious adverse reactions such as nausea and vomiting, vision loss caused by irreversible cataract, hair loss, skin diseases (such as dryness, itching, peeling and "fish scale" texture) and accelerated atherosclerosis (L98-100).
3) L96: U18666A is not a selective inhibitor of DHCR24
Thank you very much for your kind reminder. U18666A is widely used to block the intracellular trafficking of cholesterol and to mimic the Niemann-Pick type-C disease, a hereditary lysosomal storage disease. U18666A blocks the traffic of free cholesterol from late endosomal compartment. The administration of U18666A at a low dosage causes the accumulation of cholesterol in the late endosomes and lysosomes. U18666A also inhibits the cholesterol biosynthesis by inhibiting oxidosqualene cyclase and desmosterol reductase at a high dose. According to our study, we believe that the U18666A has the function to inhibit the activity of DHCR24 in our experimental system. Actually, it is widely used and accepted as the DHCR24 inhibitor such as our previous study (doi:10.7150/ijbs.63512) and the work performed by the other research groups (doi:10.1016/j.neulet.2019.05.019, doi:10.1161/STROKEAHA.115.010810). In fact, U18666A is a common drug to block DHCR24 activity through an allosteric inhibiting mechanism (doi:10.1007/s00894-016-2907-2).
4) L79-106: The article of Nes should be mentioned (doi.org/10.1021/cr200021m)
We appreciate your important suggestions. Dr.Nes is a well-known scholar in the field of cholesterol synthesis. We have read and referred to this paper (L78-81).
5) L81: The favoured biosynthesis is cell type and tissue specific
Thank you for your comments and we agree with your opinion. In the case of cholesterol biosynthesis from lanosterol, two intersecting routes have been postulated. The choice of pathway depends on the stage when the double bond on the sterol side chain at C24 is reduced. If the reduction of C24 double bond is retained until the last reaction, cholesterol synthesis proceeds via cholesta-5,24-dienol (desmosterol) (Bloch pathway). On the other hand, early Δ24-reduction involving lanosterol can proceed to cholesta-5,7-dienol (7-dehydrocholesterol) and cholesterol (Kandutsch–Russell pathway). Liver is the most important place for cholesterol synthesis, followed by the small intestine. Regardless of tissue specificity, the kinetically favored pathway for cholesterol biosynthesis appears to involve the Kandutsch–Russell pathway (L81-84).
6) L120-206: Why was FAD used as a co-factor? In literature (see also keg pathway and reactome) NADP is used? Recommend on this! I also suggest to use a reference inhibitor as a prove concept. Perhaps one compound from Ned Porter et al. or triparnol. I assume the binding of steroidal DHCR24 inhibitors is different to non-sterol inhibitors. What about verapamil, zolpidem, …
Thanks for your questions. DHCR24 enzyme contains a conserved domain (including amino acid residues 55 – 235 and 492 – 517) that participates in non-covalent flavin adenine dinucleotide (FAD) binding, which is a characteristic of a well-defined FAD-dependent oxidoreductase family (doi:10.1016/j.steroids.2008.02.007). DHCR24 can play a catalytic role after binding to FAD (doi:10.1016/j.steroids.2008.02.007). NADP is also an important substrate in the process of cholesterol synthesis, proved by the study about the activity of DHCR24 in vitro (doi:10.1086/323473). We added it to the system of cholesterol synthesis in vitro.
We appreciate your suggestion very much. As mentioned above, we used U18666A as a reference inhibitor and compared it with the new candidate inhibitor screened in the Drugbank database. In further research, we should try to select more reference inhibitors to search for candidate inhibitors, which will play a great role in our research. We added the thinking on this aspect to the discussion section (L389-393).
7) L208-225: Is the staining with Filipin highly selective? I can imagine that also desmosterol could be stained. Please provide references or experiments or even U18666.
Thanks for your questions. Filipin is the general name of a class of polyene antibiotics or polyene macrolides with four isomers. Filipin is used to combine with all free cholesterol specifically to form an aggregate or complex to produce blue fluorescence. Filipin staining is an authoritative and classic technical method for detecting total cholesterol. The technology has been meticulously improved and proved by successful experiments. It is mainly applicable to the detection of total cholesterol distribution in various cells (animals, human bodies, etc.) or cultured cells. Actually, we agree with you that Filipin can also stain the desmosterol according to its principle. However, we believe that the cholesterol synthesis pathway was totally inhibited by the treatment of DHCR24 inhibitor while the desmosterol might be accumulated a little if compared to the control. So the total sterol content was observed to be diminished demonstrated by the Filipin staining. And to further confirm the specific inhibitory effect of candidates on DHCR24 activity, we also performed experiments about the enzymatic activity assay of DHCR24 in vitro.
8) L214: I would suggest to use another inhibitor U18666 is a dirty drug.
We appreciate your suggestion. As you mentioned, U18666A has different functions. However, it is also a common drug to block DHCR24 activity through an allosteric inhibiting mechanism, especially when the study is about the DHCR24 activity (doi: 10.1007/s00894-016-2907-2). And we used the U18666A in our experimental system to fucus on its function inhibiting the DHCR24 activity with its high dosage. We believe that the influence of the other side effects of U18666A in our study about the DHCR24 activity was not so big.
9) L376-378: It’s too speculative, especially that irbesartan can be used for cancer treatment. The IC50 value of 600 nM on the isolated enzyme is quite high. I assume there are more potent inhibitors available.
We are sorry for the misunderstanding caused by this sentence. In tumor metabolism reprogramming, cholesterol metabolism reprogramming plays an important role in tumor genesis and development, mainly manifested by the up-regulation of cholesterol synthesis level in tumor cells and the abnormal aggregation of most metabolites. For example, the analysis results of The Cancer Genome Atlas (TCGA) database showed that the activity of 7 genes related to cholesterol synthesis increased in 4 tumor tissues, including melanoma, and was associated with poor prognosis (doi: 10.1158/0008-5472.CAN-15-2613). Therefore, we hope that the inhibitory effect of irbesartan on DHCR24 can not only become a candidate for the treatment of hyperlipidemia, but also be considered in the treatment of tumors. We have changed this sentence to " To summarize, our experimental results proved that drug repurposing is desirable in drug development and irbesartan might be considered a new drug for hyperlipidemia" (L394-396).
We also believe that there may be better inhibitors in the future, but according to our screening results of drugs listed in the approved drugs molecule database (Drugbank), irbesartan is the drug with the best comprehensive inhibition effect. At present, no inhibitor of DHCR24 has been used clinically. The approved drugs we screened have been used clinically for many years, so irbesartan may be a good choice in terms of development speed and cost.
10) L473-477: The used method should be described in more detail or a reference should be given. Which column was used, mobile phase, temperature, injection volume, …
We apologize for not being able to describe the experimental conditions of HPLC in sufficient detail. C18 is used as chromatographic column and methanol (chromatographic grade) is used as mobile phase at 25℃. We added detailed experimental conditions to the manuscript (L498-499).
11) L478-489: What dosage was used
We are sorry that there is no clear description of the dosage in the original manuscript. The dose we used for mice comes from the references and the exploration of the pre-experiment. Each group of mice were treated with 1% CMC-Na, simvastatin (20mg/kg), irbesartan (6mg/kg), risperidone (0.7mg/kg), tolvaptan (1mg/kg) and conivaptan (1mg/kg) (doi:10.1016/j.fct.2011.08.027, doi:10.1016/j.brainresbull.2006.03.017, doi:10.1371/journal.pone). We have added the dosage to the revised manuscript (L506-509).
12)L497:“We previously constructed …” reference is missing
Thank you for your kind reminder. We have added the missing reference (doi:10.1210/en.2008-0024) to the revised manuscript (L521-522).
13) L502-505: “were then extracted” which procedure was used?
We appreciate your kind reminder. We have supplemented the experimental steps in more detail and modified the original text to " The cell lysate was centrifuged at 15000 rpm and the protein in the supernatant was collected. The proteins from each group were extracted for further analysis" (L525-529).
Minor remarks:
L34: “hypertension and hyperlipidemia” it is speculative
Thank you for your kind reminder. Hypertension and dyslipidemia are generally considered as risk factors of cardiovascular disease (doi:10.1055/2004-869593). The analysis of cardiovascular incidence rate related to changes in blood pressure and serum cholesterol levels shows that it is very important to reduce the incidence rate by jointly reducing these two risk factors. Importantly, according to the results of our screening and validation, irbesartan is the drug with the most DHCR24 inhibition potential, and irbesartan itself is a drug used for the treatment of hypertension clinically. In addition, clinical data showed that irbesartan can reduce TC, TG, and LDL-C and increase HDL-C in patients with both hypertension and hypercholesterolemia without an exact mechanism (doi: 10.1016/j.clinthera.2016.09.005). Therefore, based on the experimental results, we believe that irbesartan may be a more suitable drug for patients with both hypertension and hyperlipidemia. In the revised manuscript, “irbesartan can be a better alternative for patients with both hypertension and hyperlipidemia” has been changed as “irbesartan may be a better alternative for patients with both hypertension and hyperlipidemia” (L370-373).
L63-64: Delete the sentence
Thank you for your suggestion. We have already deleted this sentence.
L104: “in vivo” I would suggest in mice because it was not determined in man
We appreciate your kind reminder. We have changed “in vivo” as “in mice” here (L110-112).
L150-169: The abbreviations should be explained when they are used the first time; L208: HepG2 cells
We appreciate your careful suggestions. We added explanations included Molecular Mechanics-Generalized Born Surface Area (MM-GBSA) and root mean square deviation (RMSD) (L163-L167). The explanation of FAD, DES, IRB and CON in the name of the complex is illustrated in Figure 2 and Table 3(L183-193). We also correct the name of HepG2 in the revised manuscript (L221).
L241: What means CMC-Na?
Carboxymethylcellulose sodium (CMC-Na) is a common animal drug delivery cosolvent. In this study, we used 1% CMC-Na aqueous solution to prepare a suspension to dissolve the drugs and gavage it to mice.
L274: rats or mice?
We apologize for the error, C57BL/6J mice were used in this study, so rats have been changed to mice.
Figure 6: Enlarge the chromatograms. What is shown by the bar around 15min? Chromatograms of references inhibitors should be added
Thank you for your kind suggestion. We have enlarged figure6C for a clearer observation. The peak appearing in the chromatogram in about 15 minutes is not cholesterol or desmosterol. We analyze that it should be a certain substrate from the cholesterol synthesis reaction system containing EDTA, DTT, NAD, NADP and FAD. The peak detected by HPLC may be one of these substances without DHCR24 catalytic reaction. In addition, the purpose of HPLC is to test the efficiency of inhibitors on DHCR24-catalyzed cholesterol synthesis, while U18666A have no obvious interference with the chromatogram of the reaction system, so we did not test its chromatograms.
L470: What means “free cholesterol and cholesterol”? Cholesterol is always free cholesterol
Thanks for your kind reminder. Cholesterol exists widely in almost all animal tissues in the form of free and esterified combination, mainly in the cell membrane in the free form. However, cholesterol esterase can hydrolyze esterified cholesterol into free cholesterol, and then filipin can combine with all free cholesterol to form aggregates or complexes to produce fluorescence. Therefore, filipin can stain free cholesterol and esterified cholesterol. We revised the manuscript “It can mark free cholesterol and esterified cholesterol on the structure of biofilm.”
Reviewer 2 Report
The manuscript “Virtual Screening of Novel 24-Dehydroxysterol Reductase (DHCR24) Inhibitors and the Biological Evaluation of Irbesartan in Cholesterol-Lowering Effect” is well written and well presented. It will be a good contribution to the field of medicinal chemistry. In my views, the manuscript should be accepted for publication after major revision.
· In section 4.1, the authors mentioned tools and servers like Ramachandran plot and Scoring software Maxsub, LGscore, and VERIFY3D, the proper reference to each tool or server is missing.
· Similarly, several tools (NAMD, Pymol) in the methodology section have missing references. It will be better to cite each tool with a proper reference.
· In section 4.1. Construction of 3D structure of DHCR24 proteins the authors stated that “NAMD2.9 393 was used to optimize the structure of DHCR24 at a 300 K constant temperature and to 394 perform molecular dynamics simulations for 100 ps.” However, they did not mention what protocol they used for the 100 ps simulation. Does it include all the steps of simulation or perform the minimization?
· In section 4.2, In the data set preparation, the authors said that 6858 3D structures were selected, but in the Virtual screening section, they said 50 small molecules were docked. For better understanding, provide the detail of how many compounds were screened, how they were reduced to 50, and what number of compounds were selected as final confirmers.
· In 4.4, Molecular dynamic simulations section, the authors said that constant pressure was used, what algorithm was used to maintain the pressure, and how much pressure was retained with ani pressure relaxation time.
· For more clear understanding, write in 4.4. Molecular dynamic simulations, at what interval the trajectory was saved from the final 30ns production simulation run.
· In the MM-GBSA calculation, the authors stated that the TIP3P water model was not included in the free energy calculation. However, in the equation that we calculate the ΔGpolar and ΔGnonpolar in the implicit solvent model, the authors use the generalized Born implicit solvent model when they exclude the TIP3P water model because the sum of all individual components makes the ΔGtotal in MM-GBSA calculations.
· In figure 1 (B), the Ramachandran plot text is bluer and not understandable. Please provide a high-resolution image.
· In figure 3, the 2D interactions are barely visible kindly increase the font size of the residues names and provide the image in high resolution.
· Kindly update the references with recent literature
Author Response
Answer to reviewer 2:
Comments and Suggestions for Authors:
The manuscript “Virtual Screening of Novel 24-Dehydroxysterol Reductase (DHCR24) Inhibitors and the Biological Evaluation of Irbesartan in Cholesterol-Lowering Effect” is well written and well presented. It will be a good contribution to the field of medicinal chemistry. In my views, the manuscript should be accepted for publication after major revision.
We appreciate your valuable suggestions represented below, and we have revised the manuscript and answered your questions carefully. We hope the quality of our revised manuscript is significantly improved according to your suggestions. Thanks again for your kind comments.
1)In section 4.1, the authors mentioned tools and servers like Ramachandran plot and Scoring software Maxsub, LGscore, and VERIFY3D, the proper reference to each tool or server is missing.
Thank you very much for your suggestion. In section 4.1, We revised the manuscript and added the relevant references. “Ramachandran plot (doi:10.1107/S0021889892009944) and Scoring software Maxsub(doi:10.1093/bioinformatics/16.9.776), LGscore(doi:10.1186/1471-2105-2-5), and VERIFY3D(doi:10.1016/s0076-6879(97)77022-8) were used to evaluate the model.” (L415-416).
2)Similarly, several tools (NAMD, Pymol) in the methodology section have missing references. It will be better to cite each tool with a proper reference.
Thank you very much for your suggestion. We also cited the references for Pymol(doi:10.1002/jcc.20289), NAMD(doi:10.1002/jcc.20289, doi:10.1063/5.0014475) and Ambertools(doi:10.1002/jcc.24417) (L420, L413, L447).
3)In section 4.1. Construction of 3D structure of DHCR24 proteins the authors stated that “NAMD2.9 was used to optimize the structure of DHCR24 at a 300 K constant temperature and to perform molecular dynamics simulations for 100 ps.” However, they did not mention what protocol they used for the 100 ps simulation. Does it include all the steps of simulation or perform the minimization?
Thank you for your kind reminder. The 3D structure of the constructed protein and the added hydrogen atom may have the problem of energy partial mismatch, so we optimized the DHCR24 structure before evaluation and docking. In the process of optimization, we first minimized the energy of the structure, and then carried out the process of 100ps simulation.
The optimized protocol is similar to that of dynamic simulation, all MD simulations were performed using the NAMD version 2.9 with the ff12SB process force field and the general AMBER force field (GAFF). Each complex was immersed in the TIP3P water box (12 Å from the solute surface) and neutralized by the addition of Na+ or Cl- ions. Initially, each system was subjected to energy minimization for 30,000 steps. Thereafter, the systems were sequentially heated up from 0 to 300.0 K. Finally, the non-constrained production simulations were performed for 100 ps in the constant temperature and constant pressure (NPT ensemble) simulation.
4)In section 4.2, In the data set preparation, the authors said that 6858 3D structures were selected, but in the Virtual screening section, they said 50 small molecules were docked. For better understanding, provide the detail of how many compounds were screened, how they were reduced to 50, and what number of compounds were selected as final confirmers.
We are sorry that we didn't explain our screening process clearly in the manuscript. In the screening part of the experiment, we used autodock vina and autodock4 software to screen twice. First, we used autodock vina to screen 50 potential DHCR24 inhibitors based on the lowest binding energy. Next, we used autodock4 and autogrid4 to conduct docking and correlation analysis on 50 potential inhibitors, and selected irbesartan, risperidone, conivaptan, tolvaptan according to the lowest binding energy and the highest populated clustering for subsequent research. We revised section 4.3 of the manuscript to clarify our screening process.
5)In 4.4, Molecular dynamic simulations section, the authors said that constant pressure was used, what algorithm was used to maintain the pressure, and how much pressure was retained with ani pressure relaxation time.
Thank you for your reminder. In the configuration file of NAMD, after setting the periodic boundary, we can set whether to conduct dynamic simulation under constant temperature and pressure. After the system gradually warms up to the set temperature (300K), there will be a corresponding undefined pressure value, and then the system will be dynamically simulated under this constant temperature and pressure system. Isothermal–isobaric simulations are handled in NAMD with an implementation of the Langevin piston algorithm (doi:10.1063/1.470648), which combines the Hoover constant-pressure equations of motion with piston fluctuations controlled by Langevin dynamics(doi:10.1063/5.0014475).
6)For more clear understanding, write in 4.4. Molecular dynamic simulations, at what interval the trajectory was saved from the final 30ns production simulation run.
We set timestep = 2fs in the configuration file of NAMD, which means that the trajectory was recorded every 2fs during the dynamic simulation. When using MM-GBSA method to calculate, according to the tutorial on Ambertools official website(http://ambermd.org/tutorials/), we set interval = 1. In other words, we collected the last 2000 snapshots of the dynamic simulation and calculated the binding free energy of ligand and receptor at the last 4ps of production stage. In order to make it easier to understand, we have improved the manuscript in 4.4 (L468-L471).
7)In the MM-GBSA calculation, the authors stated that the TIP3P water model was not included in the free energy calculation. However, in the equation that we calculate the ΔGpolar and ΔGnonpolar in the implicit solvent model, the authors use the generalized Born implicit solvent model when they exclude the TIP3P water model because the sum of all individual components makes the ΔGtotal in MM-GBSA calculations.
We thank you for your comments. Before MM-GBSA calculation, we prepared four topological files(prmtop) of solvated complex, complex, receptor and ligand and a trace file of dynamic simulation. However, the results only output the energy results of complex, receptor and ligand, and we calculated the binding energy by calculating the equation (EnergyBinding = EnergyComplex – EnergyReceptor – EnergyLigand). The solvated complex has not been calculated and output as a result, so the energy of the water described in our manuscript has not been calculated.
8)In figure 1 (B), the Ramachandran plot text is bluer and not understandable. Please provide a high-resolution image.
We appreciate your suggestion. We have replaced figure1B with a clearer image.
9)In figure 3, the 2D interactions are barely visible kindly increase the font size of the residues names and provide the image in high resolution.
We appreciate your suggestion. We have improved and uploaded high-resolution images.
10)Kindly update the references with recent literature
Thank you for your kind suggestions. We have reviewed and updated our references. For example: reference (doi:10.1161/01.ATV.0000131264.66417.d5, doi:10.1016/b978-0-12-024914-5.50008-5) has been replaced with reference (doi: 10.3390/cells10082007, doi: 10.1016/j.tcm.2019.01.001).
Round 2
Reviewer 1 Report
The authors present an improved work. From my point of view, a few things are still unclear.
Was SH-42 also tested in the assay? Reference substances should also be tested in the assay., e.g. SH42 is available from Cayman Chemicals. If the 42 nM was determined by yourself, there is no need for a citation behind it. If the 42 nM comes from the literature, the value is not correct! It is 4 nM, you should then also insert a citation.
Overall, pay attention to spaces after numbers.
Author Response
Answer to reviewer1:
Comments:The authors present an improved work. From my point of view, a few things are still unclear.
Was SH-42 also tested in the assay? Reference substances should also be tested in the assay., e.g. SH42 is available from Cayman Chemicals. If the 42 nM was determined by yourself, there is no need for a citation behind it. If the 42 nM comes from the literature, the value is not correct! It is 4 nM, you should then also insert a citation.
Overall, pay attention to spaces after numbers.
Thank you very much for pointing out our carelessness. In this study, we did not test the IC50 value of SH-42 to DHCR24 in the cholesterol synthesis system we built. The IC50 value of SH-42 was described from reference that the IC50 value of SH-42 is 4.2 nM. Therefore, we have revised the IC50 value of SH-42 in the manuscript and cited reference to the revised manuscript (L391).
Thanks again for your reminder. We have checked the spaces after the numbers and revised them in the manuscript (L507-508, L562).
Reviewer 2 Report
Accepted
Author Response
Thank you for your approval of our manuscript.